

# The correlation between heart rate variability index and vulnerability prognosis in patients with acute decompensated heart failure

Hongbo Liu[1], Xiaotong Wang[2] and Xiaowei Wang[3]

[1] Department of Cadre Health, Qingdao Municipal Hospital (West district), Qingdao, China
[2] Department of Cardiology, Liaocheng Third People's Hospital, Liaocheng, China
[3] Department of Traditional Chinese Medicine, Qingdao Municipal Hospital (West district), Qingdao, China

## ABSTRACT

**Objective:** To explore the correlation between Heart Rate Variability Index (HRV) and poor prognosis in patients with acute decompensated heart failure (ADHF).
**Methods:** A retrospective compilation of clinical data encompassed 128 cases of patients afflicted with acute decompensated heart failure (ADHF) who were admitted to and discharged from our hospital between April 2019 and July 2022. Subsequent to assessing their follow-up progress during the tracking period, the subjects were categorized into two cohorts: the poor prognosis group ($n = 31$) and the good prognosis group ($n = 97$). Comparative analysis of clinical data and Heart Rate Variability (HRV) parameters was executed between these two groups. Moreover, a multiple linear regression analysis was employed to identify the contributing factors associated with adverse prognoses in ADHF patients. Furthermore, the receiver operating characteristic (ROC) curve was employed to evaluate the prognostic predictive capability of HRV parameters among ADHF patients.
**Results:** The levels of SDNN (t = 3.924, $P < 0.001$), SDANN (t = 4.520, $P < 0.001$) and LF (t = 2.676, $P = 0.018$) in the poor prognosis group were significantly higher than those in the good prognosis group, and the differences were statistically significant ($P < 0.05$). The levels of PNN50 (t = 2.132, $P = 0.035$), HF (t = 11.781, $P < 0.001$) and LF/HF (t = 11.056, $P < 0.001$) in the poor prognosis group were significantly lower than those in the good prognosis group ($P < 0.05$). The results of multiple linear regression analysis indicated that SDNN, SDANN, LF, PNN50, and HF were factors influencing poor prognosis in ADHF patients ($P < 0.05$). The results of the ROC curve analysis indicate that the area under the curve (AUC) for predicting poor prognosis in ADHF patients using HRV parameters were as follows: SDNN (AUC = 0.818, 95% CI [0.722–0.914]), SDANN (AUC = 0.684, 95% CI [0.551–0.816]), PNN50 (AUC = 0.754, 95% CI [0.611–0.841]), LF/HF (AUC = 0.787, 95% CI [0.679–0.896]), and combined diagnosis (AUC = 0.901, 95% CI [0.832–0.970]). Among these, the combined diagnosis exhibited the highest AUC, sensitivity, and specificity for predicting poor prognosis in ADHF patients ($P < 0.001$).
**Conclusion:** The HRV parameters of SDNN, SDANN, PNN50 and LF/HF are closely related to the prognosis of ADHF patients. The combined detection of the

Corresponding author
Xiaowei Wang,
17669681397@163.com

above HRV parameters can improve the efficacy of predicting the poor prognosis of ADHF patients. This suggests that clinical staff can identify ADHF patients at risk of poor prognosis by long-term monitoring of HRV in the future.

## INTRODUCTION

Acute decompensated heart failure (ADHF) pertains to a clinical syndrome marked by rapid and severe cardiac functional impairment, manifesting within a brief timeframe. This condition results in an insufficient cardiac output to adequately sustain the body's metabolic demands, ultimately causing inadequate systemic tissue and organ perfusion. Clinical manifestations of ADHF often include symptoms such as dyspnea, palpitations, chest tightness, edema, fatigue, and cough, with severe cases potentially leading to shock and posing a critical threat to patient's life (*Njoroge & Teerlink, 2021*). With the increasing incidence of chronic heart failure, the occurrence of ADHF has also been on the rise (*Emmons-Bell, Johnson & Roth, 2022*). Early identification and assessment of the disease are of paramount importance for improving patient outcomes. Presently, prognostic stratification techniques for prognostic assessment in Acute Decompensated Heart Failure (ADHF) predominantly encompass biomarkers and scoring models. Among these, B-type natriuretic peptide (BNP) and N-terminal pro-B-type natriuretic peptide (NT-proBNP) stand as the prevailing biomarkers applied in the prognostic assessment of individuals afflicted with heart failure. However, BNP and NT-proBNP levels can be influenced by renal function, and their predictive value for ADHF patient prognosis requires further research and validation (*Hendricks et al., 2022*; *Pandhi et al., 2022*). Scoring models such as the ADHERE predictive model and the Seattle Heart Failure Model have limitations due to various factors, resulting in low predictive accuracy and sensitivity (*Kadoglou et al., 2022*). The Heart Rate Variability Index (HRV) reflects variations in the heartbeat interval (R-R interval) and serves as an indicator of autonomic nervous system regulation of cardiac function, encompassing both sympathetic and parasympathetic regulation. Numerous studies have demonstrated a close association between HRV and cardiac health as well as prognosis (*Tang et al., 2020*; *Hayano & Yuda, 2021*). Nevertheless, there is currently a paucity of literature investigating the correlation between Heart Rate Variability (HRV) and prognostic outcomes among patients with ADHF. This study primarily aims to investigate the connection between HRV and prognosis in ADHF patients, aiming to provide a reference basis for prognostic assessment. It is helpful for clinical staff to further identify ADHF patients at risk of poor prognosis through long-term monitoring of HRV.

## METHODS

### Study subjects

A retrospective analysis was performed on the clinical records of 128 patients diagnosed with ADHF, who were admitted to our medical institution and subsequently discharged alive during the period spanning from April 2019 to July 2022. All samples obtained in this study were approved by the ethics committee of the West Distrivt of Qingdao Municipal Hospital Group and abided by the ethical guidelines of the Declaration of Helsinki, and ethics committee agreed to waive informed consent.

(1) Inclusion criteria: Confirmed diagnosis of ADHF; age >18 years; acceptance of follow-up and completeness of case information.
(2) Exclusion criteria: Severe non-cardiac complications such as malignant tumors; concurrent systemic infections or immune dysfunction; loss to follow-up or incomplete follow-up results.

All patients underwent outpatient follow-up visits and telephone follow-up for 12 months after discharge. Patients experiencing adverse events during the follow-up period (including all-cause mortality and readmission due to recurrent ADHF) were terminated from the follow-up. Based on the follow-up outcomes, the study population was divided into a poor prognosis group ($n = 31$) and a good prognosis group ($n = 97$).

### Data collection

Patient clinical data were collected, including (1) general information: age, gender, body mass index (BMI), smoking history, alcohol consumption history, history of hypertension, history of diabetes, medication usage, length of hospital stay, *etc.*; (2) laboratory indicators: laboratory indicators such as total cholesterol, triglycerides, and urea nitrogen at the time of patient discharge were collected; (3) echocardiography: Echocardiographic results including left ventricular end-diastolic diameter (LVEDd), left ventricular posterior wall thickness (LVPWT), left ventricular ejection fraction (LVEF), *etc.*, at the time of patient discharge were collected.

### HRV measurement and criteria

#### HRV measurement method

HRV data were collected before patient discharge using the Seer Light Holter monitoring system from GE Healthcare (Chicago, IL, USA) for a 24-h three-lead dynamic electrocardiographic examination. This involved applying seven electrodes to the patient's chest for continuous 24-h monitoring. Data collected from Holter monitoring were then statistically analyzed by our institution's dynamic electrocardiography department professionals using the MARS ECG analysis system from GE Healthcare. The analysis included filtering ectopic beats and interference, excluding intervals exceeding 20% of adjacent RR intervals automatically, and considering only effective signals and sinus rhythm intervals exceeding 22 h.

*HRV assessment criteria*

Time-domain analysis (TTDA) was utilized for HRV analysis. The criteria for time-domain assessment included: (1) SDNN: Standard deviation of all normal sinus NN intervals; (2) SDANN: Dividing the entire recording into consecutive 5-minute segments, calculating the average NN interval for each 5-minute segment, then calculating the standard deviation of the average for the entire dataset; (3) PNN50: The number of heartbeats with NN interval differences greater than 50 ms divided by the total number of NN intervals, multiplied by 100. Frequency-domain indices included high-frequency power (HF) and low-frequency power (LF).

## Statistical methods

The data were subjected to analysis using SPSS version 23.0 (SPSS Inc., Chicago, IL, USA). Continuous variables that exhibited a normal distribution were presented as mean values along with their corresponding standard deviations, and inter-group comparisons were conducted through $t$-tests. Categorical data were represented as frequencies or percentages, and inter-group comparisons were carried out using chi-square tests. Multiple linear regression analysis was employed to identify the factors exerting influence on the poor prognosis in patients with ADHF. Receiver operating characteristic (ROC) curves were employed to evaluate the prognostic potential of HRV parameters in ADHF patients. A significance threshold of $P < 0.05$ was considered indicative of statistical significance.

# RESULTS

## Comparison of general characteristics between two groups

Comparison of general characteristics such as age and gender between the poor prognosis group and the good prognosis group revealed no statistically significant differences ($P > 0.05$), as shown in Table 1.

Comparison of general characteristics including age and gender between the group with poor prognosis and the group with good prognosis did not show statistically significant distinctions ($P > 0.05$), as indicated in Table 1.

## Comparison of laboratory indicators and echocardiographic parameters between the two groups

Comparison of laboratory parameters such as total cholesterol and triglycerides between the poor prognosis group and the good prognosis group exhibited no statistically significant disparities ($P > 0.05$). Moreover, echocardiographic indices encompassing left ventricular end-diastolic diameter (LVEDd), left ventricular posterior wall thickness (LVPWT), and left ventricular ejection fraction (LVEF) also showed no statistically significant differences ($P > 0.05$) between the two groups, as depicted in Table 2.

## Comparison of HRV parameters between the two groups

The poor prognosis group exhibited significantly higher levels of SDNN, SDANN, and LF compared to the good prognosis group, with statistically significant differences ($P < 0.05$). Conversely, the poor prognosis group demonstrated significantly lower levels of PNN50,

**Table 1 Comparison of general characteristics between two groups.**

| Variables | Poor prognosis group ($n = 31$) | Good prognosis group ($n = 97$) | $t/\chi^2$ | $P$ |
|---|---|---|---|---|
| Age (years, $\bar{x} \pm s$) | 71.27 ± 10.68 | 67.65 ± 9.84 | 1.746 | 0.083 |
| Gender (cases) | | | 0.029 | 0.865 |
| Male | 18 | 58 | | |
| Female | 13 | 39 | | |
| BMI (kg/m$^2$, $\bar{x} \pm s$) | 22.38 ± 1.64 | 22.19 ± 1.72 | 0.541 | 0.589 |
| Smoking history (cases) | 4 | 11 | 0.055 | 0.814 |
| Alcohol consumption history (cases) | 9 | 21 | 0.714 | 0.398 |
| History of hypertension (cases) | 12 | 27 | 1.311 | 0.252 |
| History of diabetes (cases) | 11 | 30 | 0.114 | 0.736 |
| Medication usage (cases) | | | | |
| Diuretics | 31 | 92 | 1.663 | 0.197 |
| β-Blockers | 22 | 65 | 0.169 | 0.681 |
| ACEIs/ARB | 19 | 64 | 0.227 | 0.634 |
| Digoxin | 1 | 3 | 0.001 | 0.970 |
| Anticoagulants | 30 | 91 | 0.398 | 0.528 |
| Hormones | 2 | 5 | 0.076 | 0.782 |
| Antiarrhythmics | 14 | 31 | 1.796 | 0.180 |
| Length of hospital stay (days, $\bar{x} \pm s$) | 14.12 ± 2.59 | 18.89 ± 4.56 | 1.428 | 0.156 |

**Table 2 Comparison of laboratory indicators and echocardiographic parameters between the two groups ($\bar{x} \pm s$).**

| Variables | Poor prognosis group ($n = 31$) | Good prognosis group ($n = 97$) | $t$ | $P$ |
|---|---|---|---|---|
| Total cholesterol (mmol/L) | 4.21 ± 0.56 | 3.97 ± 0.74 | 1.659 | 0.100 |
| Triglycerides (mmol/L) | 1.29 ± 0.45 | 1.32 ± 0.51 | 0.293 | 0.770 |
| Blood urea nitrogen | 11.25 ± 2.94 | 10.19 ± 3.32 | 1.589 | 0.115 |
| LVEDd (cm) | 5.72 ± 0.89 | 5.65 ± 0.77 | 0.424 | 0.672 |
| LVPWT (cm) | 0.93 ± 0.06 | 0.91 ± 0.08 | 1.280 | 0.203 |
| LVEF (%) | 45.24 ± 11.23 | 46.61 ± 11.57 | 0.578 | 0.564 |

HF, and LF/HF compared to the good prognosis group, also with statistically significant differences ($P < 0.05$). Please refer to Table 3 for details.

## Multivariate logistic regression analysis of factors influencing poor prognosis in ADHF patients

A multivariate logistic regression analysis was conducted with poor prognosis in ADHF patients as the dependent variable and significant single-factor analysis variables as independent variables. The results indicated that SDNN, SDANN, PNN50, and LF/HF significantly influenced poor prognosis in ADHF patients ($P < 0.05$). Please refer to Table 4 for details.

**Table 3 Comparison of HRV parameters between the two groups (x ± s).**

| Parameters | Poor prognosis group (*n* = 31) | Good prognosis group (*n* = 97) | *t* | *P* |
|---|---|---|---|---|
| SDNN (ms) | 139.59 ± 26.22 | 115.65 ± 30.54 | 3.924 | <0.001 |
| SDANN (ms) | 118.15 ± 16.82 | 103.63 ± 15.16 | 4.520 | <0.001 |
| PNN50 (%) | 9.65 ± 1.15 | 10.49 ± 1.62 | 2.676 | 0.018 |
| HF (ms2) | 125.41 ± 25.87 | 132.54 ± 11.65 | 2.132 | 0.035 |
| LF (ms2) | 449.52 ± 12.57 | 401.39 ± 21.57 | 11.781 | <0.001 |
| LF/HF | 1.89 ± 0.41 | 1.09 ± 0.33 | 11.056 | <0.001 |

**Table 4 Multivariate logistic regression analysis of factors influencing poor prognosis in ADHF patients.**

| Variables | Partial regression coefficient | Standard error | Standard regression coefficient | *t* | *P* |
|---|---|---|---|---|---|
| Constant | 81.257 | 12.334 | – | 6.892 | <0.001 |
| SDNN | 1.458 | 0.227 | 0.215 | 2.318 | <0.001 |
| SDANN | 1.698 | 0.295 | 0.263 | 2.411 | <0.001 |
| PNN50 | −2.254 | 0.331 | −0.368 | −5.521 | <0.001 |
| LF/HF | 1.644 | 0.279 | 0.252 | 2.241 | <0.001 |

**Table 5 ROC Curve analysis of HRV parameters predicting poor prognosis in ADHF patients.**

| HRV measure | AUC | Cut-off value | Sensitivity | Specificity | 95% CI | *P* |
|---|---|---|---|---|---|---|
| SDNN | 0.818 | 66 | 86.67 | 70.12 | [0.722–0.914] | <0.001 |
| SDANN | 0.684 | 61 | 80.15 | 70.12 | [0.551–0.816] | 0.008 |
| PNN50 | 0.754 | 8 | 82.85 | 67.69 | [0.611–0.841] | <0.001 |
| LF/HF | 0.787 | 1.5 | 80.24 | 72.19 | [0.679–0.896] | <0.001 |
| Combined diagnosis | 0.901 | | 88.97 | 83.25 | [0.832–0.970] | <0.001 |

## ROC curve analysis of HRV parameters predicting poor prognosis in ADHF patients

The outcomes of the ROC curve analysis revealed the following values for the areas under the curve (AUC) in predicting the poor prognosis of ADHF patients: SDNN, SDANN, PNN50, LF/HF, and combined diagnosis had AUC values of 0.818 (95% CI [0.722–0.914]), 0.684 (95% CI [0.551–0.816]), 0.754 (95% CI [0.611–0.841]), 0.787 (95% CI [0.679–0.896]), and 0.901 (95% CI [0.832–0.970]), respectively. The AUC, sensitivity and specificity of combined diagnosis in predicting the prognosis of ADHF patients were the highest ($P < 0.001$), as shown in Table 5 and Fig. 1.

## DISCUSSION

In this study, the clinical data of 128 patients with ADHF were collected and followed up. During the follow-up period, 31 patients with poor prognosis were found, accounting for 24.22%, which was consistent with the results reported by *Huang et al. (2022)*, indicating

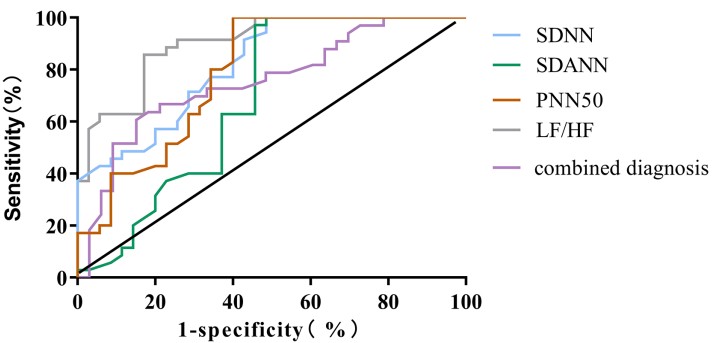

**Figure 1 ROC curve for predicting poor prognosis in ADHF patients using HRV parameters.**

that clinical attention should be paid to the prognosis of ADHF patients. ADHF is a clinical syndrome caused by severe dysfunction of the heart in a short period of time, which leads to insufficient blood perfusion of systemic tissues and organs. It often has a poor prognosis and poses a serious threat to the life safety of patients Therefore, it is of great significance to find the prognostic markers of ADHF to improve the quality of life of patients. In recent years, numerous biological markers have been progressively studied and applied in ADHF, such as B-type natriuretic peptide (BNP), N-terminal pro-BNP (NT-proBNP), mid-regional pro-atrial natriuretic peptide (MR-proANP), and cystatin C. Among these, NT-proBNP is commonly used as a gold standard for diagnosing heart failure. However, its predictive accuracy for ADHF patient prognosis can be influenced by renal metabolism (*Harrington et al., 2022*; *Núñez et al., 2022*).

Heart rate variability (HRV) reflects the regulation of the cardiac sinus node by the autonomic nervous system, indicating differences and fluctuations between successive heartbeats. It correlates with the activity of the autonomic nervous system and indicates the balance between sympathetic and parasympathetic nervous system activity. HRV can assess cardiovascular disease status and development and serves as an important indicator for predicting arrhythmias and sudden cardiac death (*Tiwari et al., 2021*). Studies have shown that reduced HRV is associated with an increased risk of cardiovascular diseases (*Cauwenberghs et al., 2021*). Lower HRV might indicate abnormal autonomic regulation of the heart and can be linked to pathological changes in the cardiovascular system. Prior studies have indicated that heightened sympathetic nervous system activity and diminished parasympathetic nervous system activity collectively contribute to the occurrence of life-threatening arrhythmias. Changes in the autonomic nervous system can be quantitatively analyzed through trends and magnitudes of heart rate variability, observed by continuous monitoring of changes in P-P intervals, P-P intervals refer to the time interval between a series of adjacent P waves on the electrocardiogram. P wave represents the atrial contraction of the heart, so the P-P interval reflects the rhythm and stability of atrial contraction (*Chen et al., 2021*).

Cardiac autonomic nerve regulation refers to the regulation of cardiac function through sympathetic and parasympathetic nerves. The autonomic nervous system is an autonomously controlled nervous system, including sympathetic and parasympathetic

nerves. Cardiac autonomic nerve regulation maintains the function and stability of the heart through the balance between excitation and inhibition of sympathetic and parasympathetic nerves (*Libbus et al., 2022*; *Wu et al., 2019*). Numerous studies confirm that ADHF patients often exhibit reduced heart rate variability, indicating autonomic regulatory dysfunction. Overactive sympathetic nervous activity can lead to vasoconstriction and inadequate myocardial oxygen supply, while insufficient parasympathetic activity can cause reduced vasodilation capacity, decreased vasodilation function will lead to increased vasoconstriction and peripheral blood flow resistance, and at the same time cause increased water and sodium retention, increased blood volume, and further aggravate the load of heart failure, thus worsening heart failure. HRV reflects the state of cardiac autonomic regulation. In a balanced state, sympathetic and parasympathetic activities are relatively equal, resulting in higher HRV. The balance of sympathetic and parasympathetic nerves is important for the proper functioning of the body. This balance can ensure the body's appropriate response to external stimuli and maintain the stability of various systems in the body. When sympathetic nerve activity is too strong or persistent, it may lead to chronic stress and stress responses, which have adverse effects on health, such as cardiovascular disease, digestive problems and decreased immune function. On the other hand, too intense or persistent parasympathetic activity may also lead to excessive relaxation of the body, affecting daily life and normal movement of bodily functions. However, an imbalance in their activities leads to reduced HRV, indicating increased sympathetic and decreased parasympathetic activity (*Konstam et al., 2022*). A study correlating HRV with the prognosis of acute coronary syndrome patients pointed out that reduced HRV may be related to factors like excessive sympathetic activity and insufficient parasympathetic activity, leading to cardiac rhythm imbalance and increased cardiac load, thereby affecting prognosis (*Pernaje Seetharam et al., 2022*). These findings suggest a strong association between HRV parameters and the prognosis of cardiovascular disease patients, indicating HRV's potential as a robust prognostic indicator.

Through single factor comparison, it was found that there was no significant difference in age, sex and other clinical data between the poor prognosis group and the good prognosis group, indicating that the above clinical data were not the factors affecting the prognosis of ADHF patients in this study. SDNN, SDANN, an PNN50 are all time-domain HRV parameters. Research by *De Couck et al. (2016)* demonstrated that SDNN significantly predicts survival time in late-stage malignancy patients, possibly mediated by reduced C-reactive protein (CRP) levels. The higher parasympathetic activity might protect cancer patients by suppressing inflammation. LF and HF are frequency-domain HRV indices, and studies indicate that LF/HF can reflect the balance between parasympathetic and sympathetic nervous activities (*Siecinski, Kostka & Tkacz, 2022*). LF represents low frequency amplitude, HF represents high frequency amplitude. *Bourdillon et al. (2022)* studies have pointed out that the heart rate variability generated by vagus nerve activity is mainly manifested as HF component, and the heart rate variability generated by sympathetic nerve activity is mainly manifested as LF component. Therefore, the change of LF/HF index may reflect the change of autonomic nervous system balance.

In this study's results, SDNN, SDANN, LF, PNN50, and HF were identified as factors adversely affecting the poor prognosis period in ADHF patients ($P < 0.05$), the results are similar to those of *Arshi et al. (2022)*, this study suggests that higher heart rate variability is associated with a higher risk of new-onset heart failure in men, highlighting a close association between HRV parameters and ADHF patient prognosis. Further analysis using ROC curves showed that SDNN, SDANN, an PNN50, LF/HF, and combined diagnosis for predicting the poor prognosis of ADHF patients had ROC values of 0.818, 0.684, 0.754, 0.787, and 0.901 respectively. Among these, combined diagnosis exhibited the highest AUC, sensitivity, and specificity, indicating the potential to enhance the efficacy of predicting poor ADHF patient prognosis through combined HRV parameter assessment. The main reasons for the combined detection of HRV parameters to predict the prognosis of ADHF patients are: (1) autonomic nerve disorder. ADHF patients are often accompanied by autonomic dysfunction. HRV can reflect the balance of sympathetic and parasympathetic nervous system, and the imbalance of this balance may affect the function and prognosis of the heart; (2) cardiovascular stability: The decrease of HRV parameters may be related to the decrease of cardiovascular stability. Lower HRV may mean decreased cardiac autonomy and impaired self-repair capacity, which may lead to poor prognosis in patients with ADHF. Additionally, routine biomarker acquisition can be cumbersome and requires periodic blood tests, having limitations. HRV parameters can be obtained using dynamic electrocardiography, which is simple, non-invasive, and doesn't cause any harm to patients. As a non-invasive technique, dynamic electrocardiography reflects changes in autonomic tension, enabling the prediction and evaluation of ADHF patients' conditions and prognosis based on HRV analysis.

## CONCLUSIONS

The HRV parameters of SDNN, SDANN, PNN50 and LF/HF are closely related to the prognosis of ADHF patients. The combined detection of the above HRV parameters can improve the efficacy of predicting the poor prognosis of ADHF patients. This suggests that clinical healthcare professionals can determine ADHF patients at risk of poor prognosis by monitoring HRV over the long term. In the future, HRV parameters can be used to analyze the specific mechanism of poor prognosis in ADHF patients.

### Funding
The authors received no funding for this work.

### Competing Interests
The authors declare that they have no competing interests.

### Author Contributions
- Hongbo Liu conceived and designed the experiments, performed the experiments, analyzed the data, authored or reviewed drafts of the article, and approved the final draft.

- Xiaotong Wang performed the experiments, analyzed the data, prepared figures and/or tables, and approved the final draft.
- Xiaowei Wang conceived and designed the experiments, performed the experiments, prepared figures and/or tables, authored or reviewed drafts of the article, and approved the final draft.

## Human Ethics

The following information was supplied relating to ethical approvals (*i.e.*, approving body and any reference numbers):

All samples obtained in this study were approved by the ethics committee of the West Distrivt of Qingdao Municipal Hospital Group and abided by the ethical guidelines of the Declaration of Helsinki.

## Data Availability

The raw data is available in the Supplemental File.

## Supplemental Information

Supplemental information for this article can be found online at http://dx.doi.org/10.7717/peerj.16377#supplemental-information.

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
