# Peer review of "The correlation between heart rate variability index and vulnerability prognosis in patients with acute decompensated heart failure"

_PeerJ, doi:10.7717/peerj.16377_

## Round 0.1 · original submission · Major Revisions

Please respond and make appropriate revisions based on the reviewers' suggestions and my comments (below). This will greatly improve the quality of the manuscript.

My comments:

1. The term "vulnerable prognosis" is unclear. It would be better to specify what is meant by this term, such as "poor prognosis" or "adverse outcomes".

2. In the Methods, the term "multifaceted linear regression analysis" is used. It would be more accurate to simply say "multiple linear regression analysis" as this is the standard terminology in statistics.

3. In the Results, it would be helpful to provide more specific information about the statistical significance of the HRV parameters. For example, instead of saying they were "notably elevated" or "significantly lower," provide the actual p-values or confidence intervals.

4. In the Conclusion, the statement that "HRV parameters are closely associated with the prognosis of ADHF patients" could be strengthened by specifying which HRV parameters were found to be most significant.

5. The abstract could benefit from a brief statement about the implications of the findings for clinical practice or future research [4].

6. The term "vulnerability assessment" is unclear. It would be better to specify what is meant by this term, such as "risk assessment" or "prognostic assessment".

7. The introduction could benefit from a clear statement of the research question or hypothesis at the end of the paragraph.

8. The introduction could provide more context about why this study is needed, such as gaps in the current literature or limitations of current methods for predicting prognosis in ADHF patients. Authors could consider citing recent related papers published in PeerJ.

9. In the ROC Curve analysis subsection, the term "vulnerable prognosis" is unclear. It would be better to specify what is meant by this term, such as "poor prognosis" or "adverse outcomes". Also, in the Discussion section, similar issues were detected.

10. The discussion could benefit from a more detailed comparison of the study's findings with those of previous research.

11. The discussion could provide more context about why this study is needed, such as gaps in the current literature or limitations of current methods for predicting prognosis in ADHF patients.

12. The conclusion could be strengthened by specifying which HRV parameters were found to be most significant.

13. [SDNN, SDANN, PNN50]: add [an] before PNN50.

Reviewer 1 ·

Basic reporting

The manuscript investigates the association between heart rate variability (HRV) parameters and prognosis in acute decompensated heart failure (ADHF) patients. The focal point of the study is the revelation that certain HRV factors could adversely affect the prognosis period and that a combined diagnosis facilitates more accurate predictions. Technically, the content expressed is mostly correct, though it would benefit from a more structured definition of HRV parameters and their relevance in the study. The introduction and background fit well into the broader spectrum of cardiac health and heart failure discussions. However, they could have been clearer in delineating the knowledge gap, which appears to be a lack of comprehensive research into the correlation between HRV parameters and ADHF prognosis. The technical standard of the methods employed is acceptable. However, elaboration of the means by which HRV parameters were obtained, the standard for categorizing 'good' and 'poor' prognosis groups, as well as the statistical soundness of the data could have improved understanding and potential replicability of the study. In conclusion, the manuscript does a satisfactory job at addressing an important aspect of cardiac health, though its narrative clarity and methodological details could be enhanced.

Experimental design

[1] The manuscript does not explain how the groups were determined.
[2] In the paragraph discussing HRV parameters, there is no clear explanation of what these parameters are. Authors should briefly describe these HRV parameters and why they are relevant in your study.

Validity of the findings

[1] The sentence "Comparison of general characteristics such as age and gender...' lacks context. It’s unclear why it’s important to compare these characteristics between the groups and what did these results convey. Authors should describe why these specific characteristics are being compared and how they might impact prognosis.
[2] The manuscript does not explain why sympathetic nervous activity and parasympathetic nervous system activity need to be balanced. Authors should give more context to why this balance is important.
[3] The ROC curve analysis results need to be discussed more thoroughly. Explain what these values indicate and why they are significant to your study.

Additional comments

[1] The introduction of heart failure is given at a much later section. Maybe, it should start with a brief introduction of heart failure, moving into its types and a focus on acute decompensated heart failure (ADHF) for better context setting.
[2] The sentence "Studies have shown that reduced HRV..." again lacks referenced sources.

Reviewer 2 ·

Basic reporting

The manuscript under scrutiny is centered around the key finding that there is a strong correlation between heart rate variability (HRV) parameters and the prognosis in patients suffering from Acute Decompensated Heart Failure (ADHF). While the veracity of the scientific content is not disputed, the language could be clearer and devoid of redundancies to make this a more comprehensive read. The introduction and background aptly address how the manuscript traverses into the larger field of cardiovascular health. However, the authors could have done a better job at demonstrating the existing knowledge void regarding the precise role of HRV parameters in ADHF prognosis and how their study attempts to fill this. On the technicality front, though the manuscript is sound, the methodologies employed could have been more detailed, and there is a vagueness about the collection of the HRV parameters. Furthermore, the integrity and robustness of the underlying data, along with control measures, could be questioned due to insufficient detail. All in all, the study makes a pertinent argument even though it suffers some issues related to clarity and methodological detail.

Experimental design

1. Please add information about how the ‘good prognosis’ and ‘poor prognosis’ groups were defined.
2. There is no definition or explanation for the terms SDNN, SDANN, and PNN50. Suggestion: Define these terms and discuss their significance.
3. The section detailing how HRV parameters were obtained lacks details. Explain more about the method used to acquire HRV parameters.

Validity of the findings

1. The discussion section jumps directly into explaining heart failure. Please transition more smoothly into this section from the results.
2. The sentence "HRV can assess cardiovascular disease status and development ..." lacks citation or supporting evidence. Provide references for this statement.
3. The effects of 'reduced vasodilation capacity' are introduced abruptly. Please explain the significance of vasodilation in the process.

Additional comments

1. In the discussion, the sentence "Dysregulation of cardiac autonomic regulation...", seems repetitive. Rephrase this sentence for better readability.
2. The discussion of LF/HF and the balance between parasympathetic and sympathetic activities lacks supporting studies. Cite relevant research.
3. The conclusion does not have any recommendation for future research. Point out potential areas where more research is recommended.

Reviewer 3 ·

Basic reporting

The author of the manuscript embarks on an exploration of the link between heart rate variability (HRV) parameters and the prognosis for patients with acute decompensated heart failure (ADHF), asserting that certain HRV parameters are potential indicators of poor prognosis. Regrettably, the paper is complicated by a dearth of clear definitions of crucial HRV parameters, which obscures the overall scientific content. The background discussion and manner in which the work addresses the broader field of cardiovascular health is convoluted. A clearer outline of the existing knowledge gap—how HRV parameters impact ADHF prognosis—and how the study intends to bridge this gap, would have been more insightful. The delineation of how HRV parameters were obtained was vague, and the methodology lacks vital detail, hindering potential replication efforts. As a whole, while the data appears robust, the control measures and statistical soundness are not elaborately explained. This study makes a valid attempt to shed light upon an essential topic, but its execution could certainly be more efficient.

Experimental design

1) It’s unclear what the 'P-P intervals' are. Provide a clearer definition.
2) The term 'cardiac autonomic regulation' is used but not explained. Briefly define cardiac autonomic regulation.
3) The sentence about SDNN predicting survival time in late-stage malignancy patients seems out of context. Explain why this detail is relevant to the study.

Validity of the findings

1) The discussion on heart rate variability (HRV) lacks sources. Support these statements with references to scientific literature.
2) The manuscript does not clearly explain how low HRV indicates abnormal autonomic heart regulation. Elaborate further on the process and mechanism involved.

Additional comments

No comment.

---

## Round 0.2 · accepted · Accept

Three reviewers approved this paper for publication. I agree with them. I think this revised article can be considered for publication in this journal, except that the following 2 issues could be addressed at the Proof stage:

1. [Objective: To explore the correlation between Heart Rate Variability Index (HRV) and poor prognosis in patients with Acute Decompensated Heart Failure (ADHF)]: Add [The aim of this study was] before [to].

2. It seems that "Comparison of general characteristics such as age and gender between the poor prognosis group and the good prognosis group revealed no statistically significant differences (P > 0.05), as shown in Table 1" is repeated twice, because they wrote, "Comparison of general characteristics including age and gender between the group with poor prognosis and the group with good prognosis did not show statistically significant distinctions (P > 0.05), as indicated in Table 1". This repetition should be removed for conciseness.

Reviewer 1 ·

Basic reporting

After the author's revisions, the basic structure of the article is clear and the language is professional.

Experimental design

I have checked and found that the author has made appropriate modifications and improvements to the content of the experimental design section, as well as provided a reasonable description of the method section.

Validity of the findings

The author has also optimized the results section well, and the data provided is reasonable.

Additional comments

I think this article has met the requirements for magazine publication and I believe it can be published.

Reviewer 2 ·

Basic reporting

The article has a clear structure and is written in professional English throughout. The references provide sufficient background information.

Experimental design

After revision, the research questions in the article have become clear, relevant, and meaningful. The described method already has sufficient detailed information for replication.

Validity of the findings

The data provided by the author is robust, statistically reliable, and controllable. The conclusion section has been fully elaborated and improved.

Additional comments

I have no other opinions.

Reviewer 3 ·

Basic reporting

I saw that the author made detailed revisions to the article and responded seriously to my comments.

Experimental design

The author made reasonable modifications to the content of the experimental design section and provided appropriate descriptions of the method section.

Validity of the findings

The author also optimized the description of the results section and provided reasonable and reliable data.

Additional comments

I believe that this article has met the requirements for magazine publication and look forward to its publication.